# Effects of Different Resistance Training Frequencies on Body Composition, Cardiometabolic Risk Factors, and Handgrip Strength in Overweight and Obese Women: A Randomized Controlled Trial

**DOI:** 10.3390/jfmk5030051

**Published:** 2020-07-17

**Authors:** Francesco Campa, Pasqualino Maietta Latessa, Gianpiero Greco, Mario Mauro, Paolo Mazzuca, Federico Spiga, Stefania Toselli

**Affiliations:** 1Department of Biomedical and Neuromotor Sciences, University of Bologna, 40126 Bologna, Italy; francesco.campa3@unibo.it (F.C.); Federico2907@gmail.com (F.S.); stefania.toselli@unibo.it (S.T.); 2Department for Life Quality Studies, University of Bologna, 47921 Rimini, Italy; pasqualino.maietta@unibo.it; 3Department of Basic Medical Sciences, Neuroscience and Sense Organs, University of Study of Bari, 70121 Bari, Italy; gianpierogreco.phd@yahoo.com; 4Unit of Internal Medicine, Diabetes and Metabolic Disease Center, Romagna Health District, 47921 Rimini, Italy; Paolo.mazzuca4@unibo.it

**Keywords:** body weight, obesity, physical activity

## Abstract

Background: Resistance training improves health in obese and overweight people. However, it is not clear what is the optimal weekly resistance training frequency and the most efficacious training protocol on body composition, cardiometabolic risk factors, and handgrip strength (HS). The aim of this study was to determine the effects of a supervised structured 24 week resistance training program on obese and overweight women. Methods: Forty-five women (BMI 37.1 ± 6.3 kg/m^2^; age 56.5 ± 0.7 years) were randomly assigned to one of two groups: A group with a high weekly training frequency of three times a week (HIGH) and a group that performed it only once a week (LOW). Cardiometabolic risk factors, anthropometric and HS measures were taken before and after the intervention period. Results: A significant (*p* < 0.05) group by time interaction was observed for body weight, body mass index, waist circumference, fat mass, plasma glucose, plasma insulin, homeostatic model assessment, and for dominant and absolute HS. Additionally, only the HIGH group saw increased HS and decreased total cholesterol and LDL-cholesterol after the intervention period (*p* < 0.05). The observed increase in HS was associated with an improved insulin resistance sensitivity (absolute handgrip strength: *r =* −0.40, *p* = 0.007; relative handgrip strength: *r =* −0.47, *p* = 0.001) after training, which constitutes an essential element for cardiovascular health. Conclusions: The results suggest that high weekly frequency training give greater benefits for weight loss and cardiometabolic risk factors improvement than a training program with a training session of once a week. Furthermore, the improvement of HS can be achieved with a high weekly frequency training.

## 1. Introduction

Overweight and obesity are major risk factors for a number of chronic diseases, including cardiovascular diseases such as heart disease and stroke, which are the leading causes of death worldwide [1]. According to the 1998 overweight and obesity clinical guidelines, overweight is defined as a BMI ranging from 25 to 29.9 kg/m^2^ and obesity as a BMI ≥ 30 kg/m^2^ [2]. From 1999–2000 to 2017–2018, the age-adjusted prevalence of obesity increased from 30.5% to 42.4%, and the prevalence of severe obesity increased from 4.7% to 9.2% in the USA [3]. In addition, the overall prevalence of obesity was similar among men and women, but the prevalence of severe obesity was higher among women and adults aged 40–59 [3]. In 2017, 52% of adults worldwide and 65% of adults in the UK were classified as overweight or obese [4]. Moreover, in most of Northern European and Southern European countries, the rate of obesity has doubled in the last 30 years [5,6].

Obesity is correlated with morphological changes resulting from an increased deposition of lipid within muscle fibers, which decreases muscle quality and contributes to frailty by reducing muscle strength and increasing disability [7]. The high percentage of overweight/obese people with the poor metabolic control of diabetes has become an important public health problem in recent years [8,9,10]. Although bariatric surgery is the most effective treatment to reduce and maintain weight loss [11,12], another effective strategy is represented by a lifestyle modification that includes a prescription for increased physical activity [13,14].

A weight change of 3% to 5% may lead to clinically meaningful health benefits and losing 5% to 10% of weight can decrease the cardiovascular risk factor, morbidity, and mortality [15,16]. Resistance training is a physical activity which promotes improvements in body composition and most cardiometabolic risk factors [17,18]. It can reduce blood pressure, total cholesterol and plasma triglycerides, inducing beneficial effects on circulating inflammatory markers and reducing insulin resistance sensitivity [19,20,21]. In addition, enhancements in body composition are associated to a better quality of life [22]. Resistance exercise is also effective to increase muscular strength [23,24], especially that of handgrip [25,26].

Recent studies, which have offered resistance training as a physical activity for obese and overweight people, have used training protocols with a frequency of two or three times a week [27,28]. Studies on training frequency in older adults have shown different results, with some reporting no differences between lower and higher frequencies on muscle mass [29,30], while others have reported that muscle strength and functional performance are favored with higher frequencies [31,32]. However, to our knowledge, no study compared the effects of different resistance training frequencies on the most informative parameters of the health status in obese and overweight people.

Therefore, the aim of this study was to compare the effects of a resistance training program of one or three times a week on body composition, cardiometabolic risk factors, and handgrip strength in overweight and obese women. We hypothesized that a higher training frequency could have a better impact on the examined parameters, especially on handgrip strength development.

## 2. Materials and Methods

### 2.1. Participants

To identify the sample size for the study, we assessed an a ‘’priori: computer required sample size- given α, power and Effect Size’’ by G*Power (3.1.9.2, Uiversität Kiel, Germany). A repeated measures ANOVA was selected as the F test of all the test family, inputing the following parameters: α = 0.05; (1−β) = 0.8; Effect Size f = 0.5. The outcomes parameters thus calculated detected a sample size of 19 participants for each group, totalling 38 participants. To meet this, we estimated a follow-up loss rate of about 10 people and selected 50 subjects who were women aged 56.5 ± 8.7 years and with a BMI 37.1 ± 6.3 kg/m^2^. They were chosen from the Lifestyle program of the ‘’Infermi’’ Endocrinology Department (Rimini) and voluntarily participated in our study. To comply with our treatment inclusion criteria, the participants could not have a body mass index ≤ 25 kg/m^2^. To prevent the affected outcomes they were not to smoke, were not to take medications and they did not have any major disease (cardiovascular disease, etc.). In addition, they had not participated in a physical exercise program over the last six months before the study. Menopausal status and menstrual cycle info were collected during two interviews before the tests, as previously stated by our group [33].

A randomized controlled trial was conducted for up to 28 weeks, of which 4 (first, second, second to last and last) were employed for assessment and measurements and 24 weeks were used for the exercise treatment. The participants were randomly allocated to each group following the study purpose: An experimental group who performed the exercise program with a weekly frequency of three times per week (HIGH) and a control group who performed the exercise program just once per week (LOW). The participants were recommended did not eat for at least two hours before the exercise program. The random allocation (random.org) was carried out by a blinded researcher. Different places were selected for measurements, which were assessed in the hospital department, and exercise which was performed in a contracted sports center. Our study followed the CONSORT statement recommendations for reporting randomized trials [32]. All participants provided their written, informed consent to participate after all procedures were explained. The study was approved by the Bioethics Committee of the University of Bologna and was conducted in accordance with the guidelines of the Declaration of Helsinki; project identification code was NCT03410329 (25 January 2018).

### 2.2. Exercise Program

The 24 week resistance training program was performed under the supervision of an exercise physiologist. During first two weeks, he exhibited how to execute each exercise correctly and theh participants had to familiarize with it. After this learning period, the exercise physiologist tested each participant workload at 10 repetitions for every exercise programmed. The resistance training program was about 65 min for both the exercise and the control group and for all of the experiment period. Exercise selection and sequence never changed over time and between groups. Workouts included an initial 10 min warm-up at low intensity (30–50% HRmax), performed using a treadmill, stepper, elliptical, or cycle ergometer, as preferred by the subjects. The central part of the program had a duration of 45 min and included seven exercises on isotonic machines with the specific sequence: leg extension and leg press for leg muscles, lat machine and low row for back muscles, pectoral machine and chest press for chest muscles and shoulder press for shoulder muscles. Following Ehrman and colleagues’ recommendations for obese people [34], every exercise provided four sets of 8–12 repetitions in a range of 60–80% one repetition maximum (1-RM); each set was performed in about 30 s with 1 min of rest between series and 2 min of rest among exercises. Whenever the participants reached less than 8 repetitions, a decreased weight of 5% was advised to perform after a short rest; similarly, whether the participants reached more than 12 repetitions, an increased weight of 5% was required to perform after a short rest. In addition, to meet the basic principles of resistance training progression [35] every 2 weeks the exercise physiologist increased the intensity by 5% on a muscular group exercise (leg, back, chest or shoulder) and tested it on the participants: if they were able to perform it, the new workload was set, whereas if they were not able to perform it, the workload did not change. Finally, the participants performed 10 min of cool down on a treadmill at the same intensity of the initial warm-up. The HIGH group performed the training program on Monday, Wednesday and Friday evening (6 p.m.) with 48 h of rest between sessions; the LOW group performed the training program on Thursday evening.

### 2.3. Anthropometry

All the measurements were carried out by a physician specifically trained in physical anthropometry and blinded to group assignment. To calculate the BMI (kg/m^2^) of each participant, we needed to evaluate the anthropometric features of height and weight. These were measured according to standard methods [36], using a stadiometer which recorded the height at an accuracy of 0.1 cm and a high-precision mechanical scale (Seca, Basel, Switzerland) for recording weight at an accuracy of 0.1 kg. Moreover, each participant’s waist circumference (WC) was recorded. It was taken at the narrowest part of the torso [36]. Skinfold thicknesses at four sites (biceps, triceps, subscapular and suprailiac) were measured to the nearest 0.1 mm with a Lange skinfold caliper (Beta technology Inc., Cambridge, MD, USA). For each anthropometrical point considered, three non-consecutive measurements were performed in order to extract the average. The technical error of measurement scores (TEM) was required to be within 5% agreement for skinfolds and within 1% for WC. Skinfold values were used in anthropometric regression equations to predict fat mass [37,38].

### 2.4. Cardiometabolic Risk Factors

Routine laboratory testing was carried out in all cases at entry; plasma glucose, total cholesterol (TC), high-density lipoprotein (HDL) cholesterol, and triglycerides (TG) were measured with routine assays using internal and external quality controls. Low-density lipoprotein (LDL) cholesterol was calculated using the Friedewald formula [39]. The homeostasis model assessment (HOMA_IR_), insulin resistance index, was calculated as the fasting plasma insulin concentration (mU/l) × fasting plasma glucose concentration (mmol/l)/22.5. Glycated hemoglobin (HbA1c) was measured by exchange high-performance liquid chromatograph using standardized laboratory procedures. Blood pressure was taken by an automatic monitor (Omron1 Model HEM-7011, Omron Healthcare, Inc., Bannockburn, IL, USA) three times successively with at least a one-minute interval between each reading. The average of the three blood pressure measurements was used for analysis.

### 2.5. Handgrip Strength

A mechanical dynamometer (Takei K.K. 5001, Takei Scientific Instruments, Ltd., Niigata City, Japan) was used to evaluate the left and right handgrip strengths at an accuracy of 0.5 kg to carry out the dominant (DHS) and absolute handgrip strength (AHS). Each participant was evaluated by keeping the dynamometer at a 90-degree flexion of their elbow, in a sitting position for a maximum of three attempts for each hand. The AHS was calculated as the sum of the right and left hand outcomes. Finally, the relative handgrip strength (RHS) was calculated following Choquette and colleagues [40], as the relationship between handgrip strength (HS) and BMI.

### 2.6. Statistical Analyses

Normality was checked with the Shapiro–Wilk test. The two-way ANOVA for between- and within-group comparisons was assessed to accept the main hypothesis. Bonferroni’s post hoc test was employed to detect specific differences in the variables where the F-ratio was significant. A two-way repeated measures ANCOVA was assessed for comparisons with the covariates as age, weight loss, and menopausal status. Effect size (*ES*) was calculated as the difference between the post-training mean and the pre-training mean related to the pooled standard deviation [8]. The *ES* was classified as follows: it was considered as small when its value was 0.20–0.49, moderate when it was 0.50–0.79, and large when it was ≥0.80. To compare the changes in weight status categories after the intervention period, the McNemar test for paired proportions was used. Bivariate and partial correlation between the changes after training in anthropometric, cardiometabolic and strength parameters were performed. Statistical significance for all the analyses was defined at *p* < 0.05. SPSS (SPSS 23.0.0.0; SPSS Inc., Chicago, IL, USA) was used for all the statistical calculations.

## 3. Results

### 3.1. Participants’ General Characteristics

The flow chart with a schematic representation of participant allocation is presented in Figure 1. Forty-five participants met the inclusion criteria: 22 women had been randomly assigned to the HIGH group and 23 to the LOW group. Of them, five subjects abandoned during the intervention period and one was excluded from analysis to prevent outlier bias. Finally, 39 participants were included in the analysis.

No significant differences in age, weight, height and BMI were found between the two groups before the intervention period (Table 1).

### 3.2. Anthropometry

There was a significant group by time interaction for weight, BMI, WC and the percentage of fat mass (%FM) with an 8.7%, 8.7%, 9.0% and 9.8% decrease in the HIGH group and with a 4.9%, 4.9%, 4.7% and 5.3% decrease in LOW group (Table 2), respectively. After adjusting for age, weight loss, and menopausal status as covariates, the group by time interaction remained significant for weight (F = 4.15, *p* = 0.049, statistical power = 0.51), WC (F = 8.02, *p* = 0.007, statistical power = 0.78) and %FM (F = 7.79, *p* = 0.008, statistical power = 0.77). Post hoc analysis revealed that in both groups all the anthropometric values significantly decreased from before to after the intervention period.

Figure 2 shows changes in the weight status categories among the participants after 24 weeks. In the HIGH group, 11 women reached a lower weight status category than at the beginning of the study (*p* = 0.001), while only seven achieved this in the LOW group (*p* = 0.01).

### 3.3. Cardiometabolic Risk Factors

A significant group by time interaction was found for plasma glucose, insulin and HOMA_IR_ (Table 2). After adjusting for age, weight loss and menopausal status as covariates, the group by time interaction remained significant for plasma glucose (F = 19.54, *p* < 0.001, statistical power = 0.99), insulin (F = 5.43, *p* = 0.025, statistical power = 0.62), and HOMA_IR_ (F = 10.83, *p* = 0.002, statistical power = 0.89). Post hoc analysis showed that, after the intervention period, the participants significantly reduced their fasting plasma glucose of 8.7% and 2.6%, insulin of 16.0% and 6.5% and HOMA_IR_ of 23% and 8.9% in the HIGH and LOW group, respectively. HbA1c, TC, TG and LDL cholesterol were significantly decreased, and the HDL cholesterol was increased in the HIGH group after the intervention period. HbA1c, TG and HDL cholesterol were similarly changed in the LOW group (*p* < 0.05). The systolic and diastolic blood pressure were not significantly changed in both groups after the 24 week intervention program.

### 3.4. Handgrip Strength

A significant group by time interaction was also found for the DHS, AHS and RHS (Table 2). After controlling for age, weight loss, and menopausal status as covariates, the group by time interaction was significant for the DHS (F = 15.09, *p* < 0.001, statistical power = 0.96), AHS (F = 15.57, *p* < 0.001, statistical power = 0.97), and the RHS (F = 23.67, *p* < 0.001, statistical power = 0.99). In the HIGH group, participants increased the strength values by 19.8%, 18.7% and 30.1% for the DHS, AHS and RHS, respectively, while for the same parameters in the LOW group the increments were 4.6%, 4.7% and 10.2%, respectively. Post hoc analyses revealed that in the HIGH group, the participants increased their DHS, AHS and RHS after the intervention (*p* < 0.05). On the contrary, the LOW group changed significantly only in RHS.

### 3.5. Correlations

The matrix of correlations between the percent variations after the Lifestyle program (Δ) in anthropometric parameters, cardiometabolic risk factors and strength values is presented in Table 3. After adjusting for age, weight loss, and menopausal status as covariates, the association remained significant between the weight and WC (r = 0.425, *p* = 0.006), %FM (r = 0.418, *p* = 0.007) glucose (r = 0.374, *p* = 0.017), and RHS (r = −0.422, *p* = 0.007). Other associations were shown between WC and %FM (r = 0.315, *p* = 0.048), HDL cholesterol (r = −0.387, *p* = 0.014), LDL cholesterol (r = 0.356, *p* = 0.024), glucose (r = 0.369, *p* = 0.019), insulin (r = 0.347, *p* = 0.028), HOMA_IR_ (r = 0.407, *p* = 0.009) and RHS (r = −0.392, *p* = 0.012), and between %FM and glucose (r = 0.495, *p* = 0.001), insulin (r = 0.380, *p* = 0.015), HOMA_IR_ (r = 0.490, *p* = 0.001) AHS (r = −0.364, *p* = 0.021) and RHS (r = −0.513, *p* < 0.001). Changes in LDL cholesterol were associated with variations in HDL cholesterol (r = −0.357, *p* = 0.024) and in HbA1c (r = 0.327, *p* = 0.040). Significantly correlations were also found among the glucose and HOMA_IR_ (r = 0.569, *p* < 0.001), AHR (r = −0.349, *p* = 0.027) and RHS (r = −0.480, *p* = 0.002), and between AHS and HbA1c (*p* = 0.039), insulin (r = −0.365, *p* = 0.021), HOMA_IR_ (r = −0.426, *p* = 0.006), DHS (r = 0.778, *p* < 0.001) and RHS (r = 0.902, *p* < 0.001). HOMA_IR_ was also correlated with insulin (r = 0.948, *p* < 0.001) and RHS (r = −0.500, *p* = 0.001).

## 4. Discussion

We found that 24 weeks of resistance training improved health in overweight and obese participants. Our resistance training program led to weight loss and a reduced BMI, improved insulin resistance and fat mass profile, and increased relative strength in participants. Aronne and Isoldi [41] suggested that changes in anthropometric and cardiometabolic risk factors are essential to improve obese health outcomes. Although some studies reported a statistically insignificant trend or no change in fat mass after a resistance training program [42,43], our study is in line with the findings of other authors who reported beneficial changes in body composition [44], fat mass and fat-free mass [45,46], muscle volume [47], physical function [48], BMI [49], and body weight [50]. According to Kraemer and Ratamess [35], the foremost principles of resistance training progression are progressive overload, specificity, and variation. We assessed these altering exercise intensity and training volume. However, one limitation may be that we never changed the exercise selection and rest over time.

We also hypothesized that the highest training frequency would have provided more positive improvements in body composition, cardiometabolic risk factors, and handgrip strength. No prior evidence was available on the differences in training frequencies in obese and overweight people. The American College of Sports Medicine [51] recommended that novice individuals should do full body resistance exercise 2–3 days per week. Our results agreed with it and showed that a higher frequency better affects obese and overweight health in women. The most important differences between the two groups relate to some of the lipid profile parameters, such as total cholesterol, LDL cholesterol, and handgrip strength. In the group that followed the resistance training program three times a week, these parameters increased significantly, unlike the group that trained once per week.

Another important result of our study was that a 24 week exercise program elicited a positive effect on metabolic syndrome risk factors such as HbA1c, HOMA_IR_ and lipid profile. A review from Lee and colleagues [19] showed that resistance exercise improved insulin resistance and lipid profile in younger obese, but the best outcomes appeared with resistance combined with aerobic exercise. In addition, Brown and associates [52] showed that a single bout of resistance exercise positively affected the glucose and insulin level during 18–24 h post exercise. However, more evidence is needed.

No significant changes in blood pressure values were measured after the intervention in both groups, though they tended towards improvements. Similar results were found by Sigal and colleagues [53], who could not find reductions in blood pressure values after 22 weeks in the participants of a resistance training program, and not even in those who practiced an aerobic training program. A limitation in our results may be due to the experimental designs that did not include diet with a caloric restriction program. Several authors found that dietary patterns combined with exercise improved obese health [54,55].

Our view is that high-frequency training is crucial in the development of strength, which also plays an important role in cardiovascular health. On this point, Lee et al. [56] reported that handgrip strength is correlated with cardiometabolic risk factors including blood pressure, triglyceride, HDL cholesterol, HbA1c, and fasting glucose. In line with that, our results highlighted significant correlations between handgrip strength and HbA1c, plasma glucose, plasma insulin and HOMA_IR_, indicating that strength development may have played a role in improving insulin resistance and cardiometabolic risk factor. Moreover, the observed increase in the handgrip strength concomitant with an improved insulin resistance sensitivity after training constituted an essential element for cardiovascular health.

Other limitations of our study included: (a) we did not consider a control group to monitor the differences of the changes in the evaluated parameters between those who practice a controlled exercise and those who do not; (b) only women were included in this experimental design, therefore our findings may not be applicable on a male sample; (c) we were not able to monitor physical activity levels outside of the study environment and to track dietary intake throughout the study, though individuals were asked to maintain their usual lifestyle habits.

Further evidence on the subject could be collected, contributing to this particular field of research and supporting the effort of clinicians working with physical exercise for obese or overweight people. The literature is abundant with resistance training studies, but very little is known regarding the effects of different frequencies of weekly training. As demonstrated by our results, resistance exercise and its weekly training frequency causes different effects on body composition, cardiometabolic risk factors and in particular on strength development. In fact, although both groups had an improvement trend after 24 weeks of intervention, the major effects on the examined parameters were measured in the group that performed the training program at a higher weekly frequency.

## 5. Conclusions

Resistance training is effective to induce improvement in body composition and cardiometabolic risk factors, but the development of handgrip strength can only be achieved with a high training frequency. This study shows that a high training frequency gave greater benefits for weight loss, cardiometabolic risk factor improvement, and handgrip strength, than a training program with a session of once a week.

## Figures and Tables

**Figure 1 jfmk-05-00051-f001:**
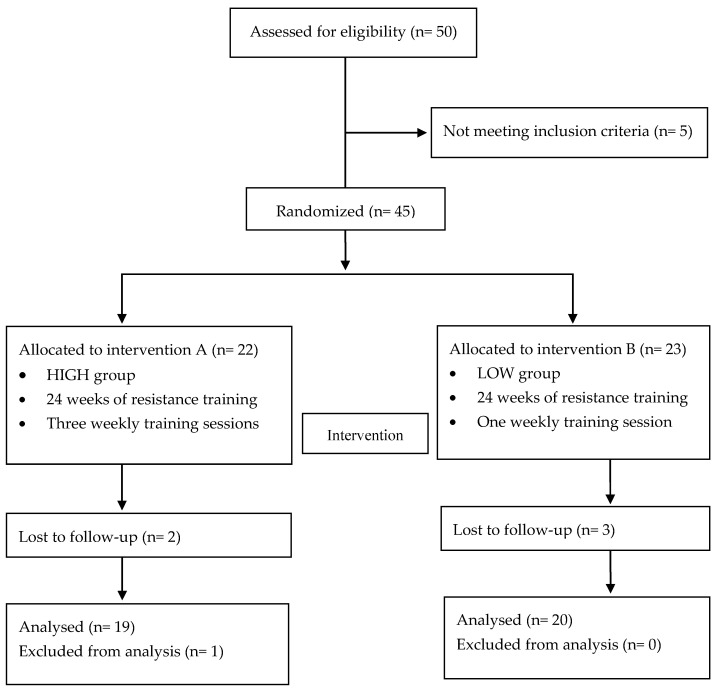
Flow chart.

**Figure 2 jfmk-05-00051-f002:**
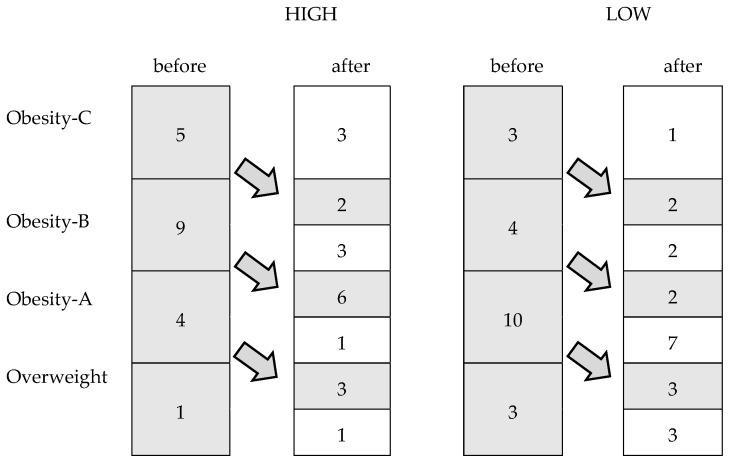
Changes in the weight status categories among the participants after the intervention period.

**Table 1 jfmk-05-00051-t001:** General characteristics of the participants before the intervention period.

Variable		HIGH (*n* = 22)	LOW (*n* = 23)	*p*
Age (years)	a	55.51 ± 9.16	57.50 ± 8.44	0.46
	b	40–69	40–68	
Weight (kg)	a	97.43 ± 17.41	88.76 ± 16.00	0.10
	b	73.50–137.10	73.20–139.30	
Height (cm)	a	158.51 ± 6.12	158.05 ± 6.60	0.81
	b	146.80–170.70	146.00–168.90	
BMI (kg/m^2^)	a	38.70 ± 6.18	35.58 ± 6.35	0.11
	b	28.10–52.10	27.50–51.00	

Note: A = mean ± SD, b = range.

**Table 2 jfmk-05-00051-t002:** Participant characteristics before and after the 24 weeks of intervention.

Variable	HIGH (*n* = 19)		LOW (*n* = 20)		*ES*	Interaction *p*-Value	SP
	Before	After	Before	After			
Anthropometry							
Weight (kg)	97.43 ± 17.41	88.73 ± 18.52 *	88.76 ± 16.00	84.02 ± 13.20 *	−0.31	<0.01	0.80
BMI (kg/m^2^)	38.70 ± 6.18	35.25 ± 6.85 *	35.58 ± 6.35	33.66 ± 5.05 *	−0.19	<0.01	0.77
WC (cm)	110.31 ± 13.68	100.49 ± 14.66 *	106.69 ± 12.26	101.46 ± 11.24 *	−0.41	<0.01	0.89
FM (%)	40.17 ± 3.29	36.26 ± 3.73 *	39.36 ± 3.36	37.24 ± 3.70 *	−5.60	<0.01	0.94
Blood pressure							
SBP (mmHg)	130.48 ±13.77	125.24 ± 17.78	127.62 ± 11.46	121.43 ± 25.93	0.04	0.89	0.05
DBP (mmHg)	81.43 ± 9.10	77.57 ± 9.66	78.81 ± 6.69	78.33 ± 10.16	0.35	0.26	0.19
Lipid profile							
TC (mg/dl)	204.50 ± 49.16	196.14 ± 44.08 *	209.85 ± 32.42	202.36 ± 27.87	−0.05	0.87	0.05
TG (mg/dl)	143.35 ± 52.90	120.18 ± 50.28 *	138.54 ± 54.25	127.69 ± 50.44 *	−0.31	0.30	0.17
HDL-C(mg/dl)	54.30 ± 15.16	59.43 ± 16.24 *	53.04 ± 13.74	54.35 ± 13.96 *	0.58	0.06	0.45
LDL-C (mg/dl)	133.58 *±* 27.76	121.29 ± 25.00 *	122.98 ± 41.62	120.33 ± 48.75	−0.60	0.05	0.58
Insulin resistance							
HbA1c (%)	6.05 ± 0.65	5.68 ± 0.67 *	6.56 ± 1.36	6.31 ± 1.22 *	−0.39	0.20	0.23
Glucose (mg/dl)	97.48 ± 13.77	88.82 ± 12.89 *	94.98 ± 7.83	92.43 ± 7.54 *	−1.33	<0.01	0.98
Insulin (u/l)	13.07 ± 2.61	11.01 ± 2.90 *	11.63 ± 2.85	10.79 ± 2.71 *	−0.76	0.01	0.68
HOMA_IR_	3.12 ± 0.68	2.40 ± 0.67 *	2.72 ± 0.69	2.45 ± 0.58 *	−1.07	<0.01	0.92
Strength							
DHS (kg)	22.76 ± 4.92	27.04 ± 5.29 *	24.50 ± 5.66	25.45 ± 5.77	1.34	<0.01	0.99
AHS (kg)	42.81 ± 9.39	50.35 ± 9.13 *	46.88 ± 11.18	48.83 ± 11.71	1.36	<0.01	0.99
RHS (kg)	1.12 ± 0.27	1.47 ± 0.36 *	1.34 ± 0.36	1.47 ± 0.39 *	1.53	<0.01	0.98

Note: Data are expressed as the mean and standard deviation. * *p* < 0.05 vs. before, BMI: body mass index, WC: waist circumference, FM: fat mass, SBP: systolic blood pressure, DBP: diastolic blood pressure, TC: total cholesterol, TG: triacylglycerol, LDL-C: low-density lipoprotein cholesterol, HbA1c: glycated hemoglobin, HOMA_IR_: homeostasis model assessment for insulin resistance, DHS: dominant handgrip strength, AHS: Absolute handgrip strength, RHS: relative handgrip strength, SP: statistical power.

**Table 3 jfmk-05-00051-t003:** Matrix of the correlations between the percentage variations of the variables after training.

	BMI	WC	FM	HDL-C	LDL-C	HbA1c	GLU	INS	HOMA_IR_	AHS	RHS
Weight	1.00 †	0.54 †	0.39 *	−0.24	0.20	0.68	0.47 †	0.29	0.35 *	−0.07	−0.45 *
BMI	1.00	0.53 †	0.38 *	−0.24	0.20	0.06	0.47 †	0.29	0.35 *	−0.06	−0.45 *
WC	-	1.00	0.29	−0.41 *	0.42 *	0.25	0.40 *	0.30 *	0.36 *	−0.21	−0.38 *
FM	-	-	1.00	−0.08	0.36 *	0.24	0.53 †	0.45 *	0.47 †	−0.39 *	−0.47 †
HDL-C	-	-	-	1.00	−0.32 *	−0.01	−0.28	−0.09	−0.16	0.10	0.18
LDL-C	-	-	-	-	1.00	0.40 *	0.21	0.20	0.23	−0.23	−0.27
HbA1c	-	-	-	-	-	1.00	0.07	−0.05	−0.07	−0.32 *	−0.29
GLU	-	-	-	-	-	-	1.00	0.41 *	0.60 †	−0.31 *	−0.47 †
INS	-	-	-	-	-	-	-	1.00	0.95 †	−0.36 *	−0.42 *
HOMA_IR_	-	-	-	-	-	-	-	-	1.00	−0.40 *	−0.47 †
AHS	-	-	-	-	-	-	-	-	-	1.00	0.88 †

Note: * *p* < 0.05, † *p* < 0.001, GLU: glucose, INS: insulin.

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
