# Peer review of "Effects of Different Resistance Training Frequencies on Body Composition, Cardiometabolic Risk Factors, and Handgrip Strength in Overweight and Obese Women: A Randomized Controlled Trial"

_jfmk, 2020, doi:10.3390/jfmk5030051_

Round 1

Reviewer 1 Report

1. The study obtained improvements but was without complying with the WHO recommendations. Embarrassingly, I never even noticed that the authors mentioned any approval by an Institutional Review Board (IRB).

2. All studies have limitations and the authors are not doing a diet-controlled study and never claimed to. Many studies in this area simply tell participants to remain as is so that the only intervention is the one being examined.

Finally, I remember stating in my review that this is not an exciting paper by checking the "average" button. However, now seeing it was not approved by an IRB, I must reject. That is unethical.

Author Response

The authors would like to thank the Reviewers for their precious and constructive advices. We found the comments very helpful in improving the paper. We have attached a copy of the Reviewers’ comments below and inserted our responses, outlining how each point has been addressed.

Reviewer 1:

The study obtained improvements but was without complying with the WHO recommendations. Embarrassingly, I never even noticed that the authors mentioned any approval by an Institutional Review Board (IRB). All studies have limitations and the authors are not doing a diet-controlled study and never claimed to. Many studies in this area simply tell participants to remain as is so that the only intervention is the one being examined. Finally, I remember stating in my review that this is not an exciting paper by checking the "average" button. However, now seeing it was not approved by an IRB, I must reject. That is unethical.

  • We have made every effort to improve the manuscript in accordance with the reviewers' comments. We are ready to edit it again if necessary. The info relating to the approval of the bioethics committee of Bologna and the registration of the trial on clinicalgov are reported in the lines: 99-101.

Reviewer 2 Report

This manuscript titled “ Effects of different resistance training frequencies on 2 body composition, cardiometabolic risk factors, and 3 handgrip strength in overweight and obese women. A randomized controlled trial” investigates the impact of different resistance training frequencies (3 d/wk vs 1 d/wk) on body composition, markers of cardiometabolic health, and handgrip strength in overweight and obese women. In general, the manuscript is clearly written and carries translational significance to the field. However, I have provided a list of comments intended to strengthen paper in good faith prior to publication.

Typology errors:

Line 22: Superscript the 2 in kg/m2.

Line 25: Should read “group by time interaction”.

Line 54: Spacing seems to be incorrect at “losing 5% to 10%”.

Line 86-87: It is good practice to avoid contractions in academic writing. I recommend changing “didn’t” and “couldn’t”.

Line 128-129: In text citation is not properly formatted.

Line 203: Comma needed before “respectively”.

Line 266: Typo on “therefore”.

Introduction:

The introduction provides sufficient background and justifies the need for this study in the introduction.

Materials and Methods:

The methods section would be strengthened with the addition of data for intraclass correlation coefficient (ICC) or coefficient of variation (CV) for quantitative measures.

If the authors are reporting that their resistance training program led to beneficial adaptations (body composition, cardiometabolic risk factors, and handgrip strength), I believe that they should elaborate on the program more. Here are my thoughts:

  • Were the subjects familiarized with the exercises used in this program? If so, how long or how many familiarizations were included? Participants lacking resistance training experience are suspect to a learning curve, which could impact the exercise execution.
  • Please be more precise on how much load was increased or decreased when subjects completed more than 10 reps or less than 8, respectively.
  • Was the exercise selection and exercise sequence the same for each session for the duration of the study? Were exercise selection and sequence identical between groups?
  • Can the author’s rationale of selecting load intensity (75% 1RM), sets (4) and rep scheme (8-12) be described/explained, or supported by previous literature? It seems logical that participants would plateau in exercise progression without altering these aforementioned variables. Was this intentional? Please explain.

The Statistical Analysis section is well written, and the data is represented well in the Results section. The description of the power analysis in the Participants section is appreciated, too. I commend the authors for their efforts here.

Discussion:

This section needs much work. Better tie in your findings with the literature and explain how the associations among body composition, lipid profiles, and insulin sensitivity/resistance influenced your findings. Particular attention needs to be paid to the cardiometabolic outcomes observed. The authors should better explain how their resistance training intervention elicited the significant changes reported in this work.

Author Response

The authors would like to thank the Reviewers for their precious and constructive advices. We found the comments very helpful in improving the paper. We have attached a copy of the Reviewers’ comments below and inserted our responses, outlining how each point has been addressed.

This manuscript titled “Effects of different resistance training frequencies on 2 body composition, cardiometabolic risk factors, and 3 handgrip strength in overweight and obese women. A randomized controlled trial” investigates the impact of different resistance training frequencies (3 d/wk vs 1 d/wk) on body composition, markers of cardiometabolic health, and handgrip strength in overweight and obese women. In general, the manuscript is clearly written and carries translational significance to the field. However, I have provided a list of comments intended to strengthen paper in good faith prior to publication.

  • Thank you for your support. We found the comments extremely helpful in improving the paper. We hope that the revised version can be an innovative contribution to the MPDI Journal.

Typology errors:

Line 22: Superscript the 2 in kg/m2.

  • Thank you for your comment. We corrected it.

Line 25: Should read “group by time interaction”.

  • Thank your suggestion. We corrected it.

Line 54: Spacing seems to be incorrect at “losing 5% to 10%”.

  • We corrected it.

Line 86-87: It is good practice to avoid contractions in academic writing. I recommend changing “didn’t” and “couldn’t”.

  • Thank you for your suggestion. We revised accordingly.

Line 128-129: In text citation is not properly formatted.

  • Thank you for underlying it. We revised the format.

Line 203: Comma needed before “respectively”.

  • We added the comma before “respectively”.

Line 266: Type on “therefore”.

  • Thank you for noting it. We corrected it.

Introduction:

The introduction provides sufficient background and justifies the need for this study in the introduction.

  • Thank you for your support. We appreciated it.

Materials and Methods:

The methods section would be strengthened with the addition of data for intraclass correlation coefficient (ICC) or coefficient of variation (CV) for quantitative measures.

  • Thank you for your comment. We reported the measurement related to the technical error of measurement to show the accuracy of anthropometric measurements.

If the authors are reporting that their resistance training program led to beneficial adaptations (body composition, cardiometabolic risk factors, and handgrip strength), I believe that they should elaborate on the program more. Here are my thoughts:

Were the subjects familiarized with the exercises used in this program? If so, how long or how many familiarizations were included? Participants lacking resistance training experience are suspect to a learning curve, which could impact the exercise execution.

  • Thank you for your comment, we retain it useful. We added more details in lines 104-105.

Please be more precise on how much load was increased or decreased when subjects completed more than 10 reps or less than 8, respectively.

  • We added more details in lines 117-119.

Was the exercise selection and exercise sequence the same for each session for the duration of the study? Were exercise selection and sequence identical between groups?

  • Thank you explained this part. We added more details in line 108.

Can the author’s rationale of selecting load intensity (75% 1RM), sets (4) and rep scheme (8-12) be described/explained, or supported by previous literature? It seems logical that participants would plateau in exercise progression without altering these afore mentioned variables. Was this intentional? Please explain.

  • Thank you for your comment. We explained the rational selection adding more details supported by previous literature in lines 113-116.

The Statistical Analysis section is well written, and the data is represented well in the Results section. The description of the power analysis in the Participants section is appreciated, too. I commend the authors for their efforts here.

  • Thank you for your support.

Discussion:

This section needs much work. Better tie in your findings with the literature and explain how the associations among body composition, lipid profiles, and insulin sensitivity/resistance influenced your findings. Particular attention needs to be paid to the cardiometabolic outcomes observed. The authors should better explain how their resistance training intervention elicited the significant changes reported in this work.

  • Thank you for your suggestions. We improved this section adding more literature outcomes related with our findings in lines 244-246, 249-252, 255-259, 264-269.

Reviewer 3 Report

The aim of this study was to compare the effects of a resistance training program of one or three times a week on body composition, cardiometabolic risk factors, and handgrip strength in overweight and obese women. Obviously, greater improvements are expected with a third of more sessions carried out by the HIGH group. Would frequency be a relevant variable with the same number of sessions?

Surprisingly, improvements were obtained even without complying with the WHO recommendations. I think that it is the most relevant of the study. However, there are several limitations.

Abstract

L22: kg/m2 should be… kg/m2

Introduction

L39 the first sentence is well-knew knowledge. Please delete or re-write.

L40 Please use WHO as reference to this information.

L52 I think that this affirmation could be the first sentences of the paragraph.

L55 I think that this sentence could be included in the next paragraph where you mention the most effective treatment.

L59 Why resistance training is particularly suitable for obese? Please, clarify and restricting this paragraph. For example, L63 physical exercise is also… physical activity in general or resistance training?

L67 Please include previous studies in overweight-obese women only.

Methods

L82, please indicate the estimated follow-up loss rate.

L84 what do you mean with “intentionally”?

L87 what about previous physical activity level and menstrual cycle? Menopausal? Diet?  How were these controlled?

L89 Please explain the intervention protocol more extensively and in order.

L93 Please include more details about assessment: hours, days, fasting, …

L101 Schedule and rest between sessions?

L102 Was the 1RM update along the program?

L103 please clarify the intensity of warm-up and cool-down.

L109, please clarify decrease and increase weight.

L122 please indicate how the percentage of fat mass were calculated?

L129 How fasting plasma insulin concentration was calculated?

L130 please it is the first time that appear, abbreviation

L147 how menopausal status was evaluated?

Results

L158 Loss rate? Reasons?

L159 please use height instead of stature

L190-193 Please re-write in two sentences, it is too extensive.

Discussion

L225 please use was…

Discussion should be vastly improved. Currently the authors repeat the results obtained and compare with other studies but do not justify the results obtained.

There are several limitations especially the physical activity and diet. These could have a great influence in these parameters.

Table 2

Please FM (%) should be F (%) based on the text.

Author Response

The authors would like to thank the Reviewers for their precious and constructive advices. We found the comments very helpful in improving the paper. We have attached a copy of the Reviewers’ comments below and inserted our responses, outlining how each point has been addressed.

The aim of this study was to compare the effects of a resistance training program of one or three times a week on body composition, cardiometabolic risk factors, and handgrip strength in overweight and obese women. Obviously, greater improvements are expected with a third of more sessions carried out by the HIGH group. Would frequency be a relevant variable with the same number of sessions?

Surprisingly, improvements were obtained even without complying with the WHO recommendations. I think that it is the most relevant of the study. However, there are several limitations.

  • Thank you for your support. We found the comments extremely helpful in improving the paper. We hope that the revised version can be an innovative contribution to the MPDI Journal.

Abstract

L22: kg/m2 should be… kg/m2

  • Thank you for your comment. We corrected it.

Introduction

L39 the first sentence is well-knew knowledge. Please delete or re-write.

  • Thank you for your suggestion. We revised lines 39-41.

L40 Please use WHO as reference to this information.

  • We added the WHO’s reference.

L52 I think that this affirmation could be the first sentences of the paragraph.

  • Thank you for your comment. We revised the first sentence in lines 49-51.

L55 I think that this sentence could be included in the next paragraph where you mention the most effective treatment.

  • Thank you for your suggestion. We agreed and moved it in line 56-57.

L59 Why resistance training is particularly suitable for obese? Please, clarify and restricting this paragraph. For example, L63 physical exercise is also… physical activity in general or resistance training?

  • Thank you for your explanation. We improved this paragraph trying to clarify resistance exercise benefits, in lines 57-62.

L67 Please include previous studies in overweight-obese women only.

  • We changed reference 26 including a previous study in overweight-obese women only, in line 65.

Materials and Methods

L82, please indicate the estimated follow-up loss rate.

  • Thank you for your comment. We reported this info in in lines 80-81.

L84 what do you mean with “intentionally”?

  • Thank you for your question. We changed it to voluntary in line 83, meaning that Infermi clinician and medical staff did not force participants decision.

L87 what about previous physical activity level and menstrual cycle? Menopausal? Diet?  How were these controlled?

  • Thank you for your comment. We improved this section by adding new info in lines 86-88.

L89 Please explain the intervention protocol more extensively and in order.

  • We improved the whole Materials and Methods section.

L93 Please include more details about assessment: hours, days, fasting, …

  • Thank you for your suggestion, we retain it useful. We added more info in lines 93, 94 and training time in lines 124-126.

L101 Schedule and rest between sessions?

  • We added more details in lines 114-116, 125, 126.

L102 Was the 1RM update along the program?

  • Thank you for your suggestion, we appreciate it. We added more details in lines 119-122.

L103 please clarify the intensity of warm-up and cool-down.

  • Thank you for your comment. We advised participants to start and end at low intensity, added in line 109.

L109, please clarify decrease and increase weight.

  • We added more details in lines 119-122.

L122 please indicate how the percentage of fat mass were calculated?

  • Fat mass percentage was calculated using the Siri’s equation: (%FM = [ (4.95/BD) – 4.5] x 100). We added this reference in line 140.

L129 How fasting plasma insulin concentration was calculated?

  • Routine tests were performed in hospital laboratories. We have reported this info online.

L130 please it is the first time that appear, abbreviation

  • Thank you for your suggestion. We exhibited its full name in lines 145-146.

L147 how menopausal status was evaluated?

  • Thank you for your comment. We reported the following info in lines 86-88: Menopause and menstrual cycle info were collected during two interviews before the tests.

Results

L158 Loss rate? Reasons?

  • Thank you for your comment. We added more details in lines 174-176.

L159 please use height instead of stature

  • We corrected it in line 178.

L190-193 Please re-write in two sentences, it is too extensive.

  • Thank you for your suggestion, we retain it makes clearer the paragraph. We modified it in lines 209-211.

Discussion

L225 please use was…

  • We changed the phrase in lines 242-244.

Discussion should be vastly improved. Currently the authors repeat the results obtained and compare with other studies but do not justify the results obtained.

  • We appreciate the reviewer comment. We try to improve the whole discussion section adding more details in lines 244-246, 250-252, 255-258, 262, 264-269.

There are several limitations especially the physical activity and diet. These could have a great influence in these parameters.

  • We understood your concern. We know that lack of diet program limits our results and explain it in lines 274-276. In addition, we added more exercise limitations in lines 252, 261-262.

Table 2

Please FM (%) should be F (%) based on the text.

  • We revised accordingly in lines 186, 190, 228, 229, 232.

Round 2

Reviewer 1 Report

I accept now that it has been approved by an IRB.

Reviewer 2 Report

Line 34: omit 'only'

Line 186: change 'respectively' to 'respective'

Line 211: insert comma after 'group'

Line 255: edit to 'No prior evidence is available...'

Line 257: 'highest' should read 'higher'

Line 261: Consider changing to 'unlike the group that trained once per week'. Just stating 'unlike the second group' seems vague. 

Line 261-262: I do not see this as a limitation as the purpose of the study was to investigate 3 d/wk (HIGH) vs 1 d/wk (LOW). 

Line 263: edit to 'Another important result of our study was that a 24 week resistance exercise program elicited positive effects on...'

Line 267: edit to ' a single bout of resistance exercise positively affected glucose....'

Line 268: edit to 'more evidence is needed'. 

Line 286: edit to 'Other limitations of our study include: '

How many weeks was the resistance exercise program? The abstract says 6-months, Line 80 says up to 28 weeks, line 263 says 24 weeks, and line 298 says 26 weeks. Please make this time period consistent through the manuscript. 

Author Response

The authors would like to thank the Reviewers for their precious and constructive advices. We found the comments very helpful in improving the paper. We have attached a copy of the Reviewers’ comments below and inserted our responses, outlining how each point has been addressed.

Line 34: omit 'only'

Line 186: change 'respectively' to 'respective'

Line 211: insert comma after 'group'

Line 255: edit to 'No prior evidence is available...'

Line 257: 'highest' should read 'higher'

Line 261: Consider changing to 'unlike the group that trained once per week'. Just stating 'unlike the second group' seems vague.

Line 261-262: I do not see this as a limitation as the purpose of the study was to investigate 3 d/wk (HIGH) vs 1 d/wk (LOW).

Line 263: edit to 'Another important result of our study was that a 24 weeks resistance exercise program elicited positive effects on...'

Line 267: edit to ' a single bout of resistance exercise positively affected glucose....'

Line 268: edit to 'more evidence is needed'.

Line 286: edit to 'Other limitations of our study include: '

How many weeks was the resistance exercise program? The abstract says 6-months, Line 80 says up to 28 weeks, line 263 says 24 weeks, and line 298 says 26 weeks. Please make this time period consistent through the manuscript.

  • Thank you for your comments and suggestions. We improved all sections and several sentences of the manuscript accordingly.

Reviewer 3 Report

Thank you so much for your work over text.

Introduction

L49 – L62 The introduction should be improved. The authors repeat constantly general knowledge and it is difficult reading. I suggest explain obesity problem, negative effect on body, treatment,…

Methods

L83, participated....

L87 what about previous physical activity level and menstrual cycle? Menopausal? Diet?  How were these controlled?

L129 Please include more details.

L147 Please include more details.

Discussion

Despite the discussion has been improved, the main reason to reject the manuscript is that, in my opinion, the results are obvious due to the total number of sessions is different between weekly frequency. I think that it is a several methodological mistake.

Author Response

The authors would like to thank the Reviewers for their precious and constructive advices. We found the comments very helpful in improving the paper. We have attached a copy of the Reviewers’ comments below and inserted our responses, outlining how each point has been addressed.

Introduction

L49 – L62 The introduction should be improved. The authors repeat constantly general knowledge and it is difficult reading. I suggest explain obesity problem, negative effect on body, treatment,…

L83, participated....

  • We have improved all the manuscript sections and clarified the meaning of different sentences following the reviewers' comments.

L87 what about previous physical activity level and menstrual cycle? Menopausal? Diet?  How were these controlled?

  • Thank you for your comment. We added more menopausal and menstrual cycle details in line 88. We explained physical activity level in lines 87-87, where we specified that participants did not participate to physical exercise program over the last six months. Finally, we exhibited that diet exclusion is a limitation of our study in lines 273-274.

L129 Please include more details.

L147 Please include more details.

  • We appreciate your suggestions. We provided more detail during the two rounds of the review process.

Discussion

Despite the discussion has been improved, the main reason to reject the manuscript is that, in my opinion, the results are obvious due to the total number of sessions is different between weekly frequency. I think that it is a several methodological mistake.

  • While understanding your concern we believe that the innovation of this study is to compare two different training frequencies, providing scientific data to support the existing literature. Also, we provide a further positive evidence concerning the resistance training in obese women.

Round 3

Reviewer 3 Report

The main reason to reject the manuscript is that, in my opinion, the results are obvious due to the total number of sessions is different between weekly frequency. I think that it is a several methodological mistake.